# The 5′-3′ Exoribonuclease XRN4 Regulates Auxin Response via the Degradation of Auxin Receptor Transcripts

**DOI:** 10.3390/genes9120638

**Published:** 2018-12-17

**Authors:** David Windels, Etienne Bucher

**Affiliations:** 1Botanical Institute, University of Basel, Zurich-Basel Plant Science Center, Part of the Swiss Plant Science Web, Schönbeinstrasse 6, 4056 Basel, Switzerland; 2IRHS, Agrocampus-Ouest, INRA, Université d’Angers, SFR 4207 QuaSaV, 49071 Beaucouzé, France; etienne.bucher@inra.fr

**Keywords:** XRN4, RNA Decay, Auxin

## Abstract

Auxin is a major hormone which plays crucial roles in instructing virtually all developmental programs of plants. Its signaling depends primarily on its perception by four partially redundant receptors of the TIR1/AFB2 clade (TAARs), which subsequently mediate the specific degradation of AUX/IAA transcriptional repressors to modulate the expression of primary auxin-responsive genes. Auxin homeostasis depends on complex regulations at the level of synthesis, conjugation, and transport. However, the mechanisms and principles involved in the homeostasis of its signaling are just starting to emerge. We report that *xrn4* mutants exhibit pleiotropic developmental defects and strong auxin hypersensitivity phenotypes. We provide compelling evidences that these phenotypes are directly caused by improper regulation of TAAR transcript degradation. We show that the cytoplasmic 5′-3′ exoribonuclease XRN4 is required for auxin response. Thus, our work identifies new targets of XRN4 and a new level of regulation for TAAR transcripts important for auxin response and for plant development.

## 1. Introduction

Plant growth, organogenesis, and diverse responses to environmental changes are control by auxins plant hormones [1,2]. The perception of Auxin signaling depends on auxin receptors of the TAAR [TRANSPORT INHIBITOR RESPONSE 1 (TIR1)/AUXIN SIGNALLING F-BOX PROTEIN 2 (AFB2)] family. There are four TAARs genes which have partially-redundant functions [3,4,5]. TAAR proteins belong to SKIP/CULLIN/F-BOX (SCF)-ubiquitin ligase complexes and in presence of auxins can form a complex with AUXIN/INDOLE-3-ACETIC ACID (AUX/IAA) transcriptional repressors [6,7,8]. These AUX/IAA proteins are ubiquitinated and degraded by the 26S proteasome [6]. The degradation of AUX/IAA induces the release of AUXIN RESPONSE FACTOR (ARF) transcription factors. ARFs leads to the activation (5 ARFs) or to the repression (18 ARFs) of primary auxin-responsive genes [1,9,10]. 

Auxin signaling is an extremely robust system which invokes complex homeostatic regulations at the levels of synthesis, conjugation, and transport [2,11]. Moreover, recent works, including ours, have shown that it also involves complex regulation at the level of signaling [12,13,14,15,16,17]. Thus, auxin signaling homeostasis also appears to depend on the homeostasis of TAAR transcripts and to involve tight regulation orchestrated by the microRNA miR393 and by a complex network of secondary small interfering RNAs (siRNAs), the siTAARs [13,14,15,18]. Importantly, this regulation is highly significant for auxin signaling, since it appears necessary for the production of proper signaling outputs important for plant development [16].

RNA surveillance and RNA decay encompass a large array of ubiquitous regulatory mechanisms involved in controlling the integrity and turn-over of cellular RNAs [19,20]. These mechanisms depend on specific recognition factors and on the activity of endonucleases and of 5′-3′ and 3′-5′ exonucleases to degrade decapped, improperly terminated, improperly processed or improperly spliced RNA molecules [19,20]. In Arabidopsis, the family of 5′-3′ exonucleases comprises 3 proteins: XRN2 and XRN3, which are localized in the nucleus and are involved in processing of pre-rRNA and for surveillance of proper termination of transcription [21,22], and XRN4, which is localized in the cytoplasm and is implicated in the degradation of a subset of 3′ RNA fragments generated upon microRNA-guided cleavages [23,24,25,26]. Importantly, none of the XRN proteins of plants seem to have a general role in mRNA turn-over [23,24,26]. Indeed, although the steady-state level of many transcripts is increased in *xrn4* mutants, none of these transcripts exhibit altered turn-over rate [27]. For example, although XRN4 is an important component of signaling by the plant hormone ethylene, its involvement in this signaling pathway, via regulation of EIN3 BINDING F-BOX1 and 2 (EBF1 and EBF2) messenger RNA (mRNA) levels, was shown to be indirect [28,29,30,31]. Still recently, Merret et al. found that many transcripts may be targeted by XRN4 under heat stress, and at least eight of those were directly degraded [32].

We report here that *xrn4* mutant plants exhibit pleiotropic developmental defects and hypersensitive responses to auxin that are caused by their inability to generate accurate auxin response. We show that *xrn4* mutants are unable to properly regulate the turn-over of TIR1/AFB2 auxin receptor (TAAR) transcripts and that chemical suppression of TAAR functions leads to suppression of certain developmental defects observed in *xrn4* plants. Thus, our data provide evidences showing that XRN4 is a component and a direct regulator of the auxin signaling pathway. Our work identifies new direct XRN4 target RNAs and a new layer of regulation of TAAR transcripts.

## 2. Materials and Methods

### 2.1. Plant Material, Growth Conditions, and Treatments

For the scoring of developmental phenotypes and assays of cotyledon epinasty, *Arabidopsis thaliana* plants were grown in soil under long day conditions (16 h 10,000 lux lights at 21 °C/8 h dark at 16 °C) in growth chambers at 65% relative humidity [15,16,33]. Assays for inhibition of root elongation were done as described in [4]. For scoring the incidence of leaf serration, seeds were germinated and grown for 28 days on MS medium supplemented with PEO-IAA (alpha-(phenylethyl-2-oxo)-IAA) in dimethyl sulfoxide (DMSO) or DMSO alone as a control in 5.5 cm high Petri dishes. For studies of IAA3pro:GUS, IAA12pro:GUS, and DR5pro:GUS expression, seven day old seedlings were incubated in MS medium containing 10 µM 2,4-D in ethanol or ethanol alone as a control for the time indicated before proceeding to staining. For studies of AXR3-NT:GUS fusion protein stability, seven day old seedlings were placed in water at 37 °C for 2 h to induce the expression of the *HSpro:AXR3-NT:GUS* gene and incubated for 30 min at room temperature before proceeding to GUS staining. For the treatments, either 10 µM 2,4-D were added to the solution 4 h before the heat shock.

### 2.2. Histochemical GUS Assays

Arabidopsis seedlings were incubated into a staining solution containing 1 mM X-Gluc in 100 mM Na_3_PO_4_ (pH 7.2), 0.1% Triton X-100, 5 mM K_3_Fe(CN)_6_, and 5 mM K_4_Fe(CN)_6_ for 24 h at 37 °C in the dark. Seedlings were then cleared in 70% ethanol for two days and mounted in 50% v/v glycerol before observations.

### 2.3. RNA Decay Assay

For determining the rate of RNA decay, seeds were grown in short day (SD) conditions for 14 days in liquid MS medium and treated with 0.6 mM cordycepin as described in [34]. Tissues were collected at the indicated intervals before RNA extraction. Half-life times of transcripts were estimated by calculation of signal decay rate per minute.

### 2.4. RNA Preparation and RNA Analysis

RNA extractions and Northern blot hybridizations were done as described in [15].

## 3. Results

### 3.1. XRN4 Defective Plants Exhibit Pleiotropic Developmental Defects

We observed that mutants impaired in the function of XRN4 exhibit developmental abnormalities which resulted in an increase of their global stature and altered the morphology of several of their organs, as for the T-DNA insertion allele *xrn4-3* (Figure 1). As exemplified in Table 1, *ein5-7* mutants, another allele of *xrn4*, develop longer leaves and generate significantly more of these leaves (Student *t*-test, *p* = 1.2 × 10^−5^) than the wild-type (wt) plants. Their rosettes are three times heavier than those of wt plants (*p* = 4.4 × 10^−7^), their stems have shorter internodes (*p* = 2.7 × 10^−6^), and they develop a greater number of flowers (*p* = 1.9 × 10^−3^). Their root system is also affected; primary roots are longer with less lateral roots (*p* > 1.6 × 10^−5^) than in wt plants. The morphology of their organs is also altered; the leaves develop significantly more serrations (*p* > 3 × 10^−4^) and the flowers are more often duplicated (*p* = 3.1 × 10^−4^) in *xrn4* compared to wt plants. Thus, deficiency in XRN4 function leads to production of pleiotropic morphological defects. These observations suggested that XRN4 regulates the expression of genes involved in several important developmental programs.

### 3.2. Defects in XRN4 Result in Hypersensitive Physiological Responses to Auxin

The pleiotropic defects observed for *xrn4* plants, especially their greater stature, their affected root architecture, and their serrated leaves suggested that these mutants are deficient in the regulation of auxin responses. This hypothesis was also supported by our observation that, in normal axenic growth conditions, *ein5-7* and *xrn4-3* mutants exhibit a significantly higher frequency of root enrolling (37.3% and 52.2%, respectively) (student *t*-test, *p* < 1.2 × 10^−5^, *n* = 66) than wild-type plants (11.9%) which was similar to that of the hypersensitive auxin mutants *mir393b-1*, and thus indicative of defects in auxin-dependent gravitropic responses (Appendix A).

To test whether auxin-dependent physiological and developmental responses are affected in *xrn4* mutants, we compared the incidence of extreme cotyledon epinasty, typical for hypersensitive auxin response [15,16] in wt, in the two *xrn4* alleles *xrn4-3* and *ein5-7*, and in the *ein3-1* mutant, which serves as a control to evaluate indirect effects of ethylene signaling deficiency (Figure 2). When grown on standard medium, a high and significantly greater fraction of *xrn4-3* and *ein5-7* mutants (>74%; Fischer exact test, *p* < 3.1 × 10^−8^) than wt (2%) or *ein3-1* (19%) plants exhibited the extreme cotyledon epinasty phenotype typical for hypersensitive response to auxin (Figure 2). Similar to what we observed earlier for miR393-deficient mutants and the siTAARs deficient mutant *dcl4-2* [15,16], this incidence was decreased by increasing the concentration of the auxin transport inhibitor NPA (1-N-naphthylphthalamic acid) in the medium. Thus, these observations showed that *xrn4* mutants exhibit a hypersensitive response to auxin and that the cotyledon epinasty response to auxin depends on the regulation by XRN4, independently of the ethylene pathway.

As an additional test of hypersensitive auxin responses, we compared the inhibition of root elongation typical for response to the synthetic auxin 2,4-D in these organs [4]. Although the changes were subtler, the root elongation of both *xrn4* mutant alleles was slightly more inhibited than that of wt plants at all 2,4-D concentrations tested (Appendix A). The 2,4-D concentration necessary to reach 50% inhibition (RI50) was 12% to 24% lower for *xrn4* alleles than for wt. Thus, these observations showed that *xrn4* mutants also exhibit an auxin hypersensitive response phenotype in roots. Together with the cotyledon epinasty phenotype, these experiments established that XRN4 is important for proper physiological and developmental auxin-mediated responses.

### 3.3. Chemical Inhibition of TAAR Function Suppresses Specific *xrn4* Phenotypes

The *xrn4* plants exhibit pleiotropic developmental defects, and at least some of these are hypersensitive auxin responses. To test whether some of the *xrn4* phenotypes are directly caused by improper auxin signaling, we suppressed the activity of TAAR proteins with the auxin antagonist PEO-IAA, which blocks auxin signaling by occupying the auxin binding site of TAAR proteins [35]. Interestingly, a significantly greater fraction of *xrn4-3* plants (76%; Fischer exact test, *p* < 1.8 × 10^−11^) relative to wt plants (12.5%) showed an extreme cotyledon epinasty phenotype which was gradually decreased by increasing the concentration of PEO-IAA in the medium (Figure 3A). Furthermore, 80 µM of PEO-IAA was necessary to observe no significative difference between the *xrn4-3* and wt plants.

We also measured the effect of PEO-IAA on the incidence of leaf serrations (Figure 3B). While the number of serrations, which we measured on the third and fourth leaf, was significantly higher in *xrn4* compared to wt plants grown on MS medium, this number was decreased to similar values when 80 µM PEO-IAA was applied.

Thus, these experiments demonstrated that the chemical suppression of TAAR activity leads to suppression of some of the developmental and physiological defects of *xrn4* mutants.

### 3.4. XRN4 Regulates the Turn-Over of TAAR Transcripts

The molecular function of XRN4, the phenotype displayed by *xrn4* plants that could be reverted by a *TAAR* inhibitor and their hypersensitive auxin responses which were reminiscent to what we previously recorded for *mir393b-1* suggested a potential role for XRN4 in the regulation of *TAAR* transcripts. Thus, to determine whether these hypersensitive auxin responses was the result of improper regulation of *TAAR* gene expression, we analyzed their steady-state mRNA levels in *xrn4* mutants by northern blot. Interestingly, we found that the levels of full-length *TIR1*, *AFB2,* and *AFB3* mRNAs was higher in *xrn4* mutants than in wt plants (Figure 4A). AFB1, as for it, could not be detected using this method. To know if the increase of *TAAR* transcripts in *xrn4* plants can affect the production of siTAARs derived from *AFB2* (AFB2 3′D2(+)) (Figure 4B), we analyzed the accumulation of AFB2 3′D2(+) in *xrn4-3* and *xrn4-3/rdr6-14*. Whereas the level of mir393 still remained unchanged in *xrn4-3* (Appendix A), we can observe an increase of AFB2 3′D2(+) in *xrn4-3* but not in the double mutants *xrn4/rdr6* as expected, because the production of AFB2 3′D2(+) is dependent of RDR6. 

To understand why the increase of siTAARs did not result in a decrease of *TAAR* transcripts, we analyzed the accumulation of *AFB2* and AFB2 3′D2(+) in the *xrn4*/*rdr6* double mutants. In the *xrn4/rdr6* double mutants the level of *AFB2* is much higher than in the *xrn4* mutants (Figure 4C), indicating that XRN4 is important for the control of TAAR transcript and siTAAR levels. The expression of auxin receptor genes of the *TIR1*/*AFB2* clade is affected in *xrn4* mutants. However, whether this effect is direct or indirect, and, whether it is responsible for the hypersensitive auxin responses of *xrn4* plants remains to be addressed.

To address if the *TIR*/*AFB2* transcripts are targeted directly or indirectly by XRN4, we treated the plants with the transcription inhibitor cordycepin [30] and we measured the kinetic of *TAAR* transcripts decay. The signals obtained by northern blot for *TIR1* and *AFB3* were too weak to provide informative conclusion, however, for *AFB2*, the signals for full-length mRNAs decreased faster in wt than in *xrn4* plants. Indeed, the half-life of *AFB2* transcripts was more than three times longer in *xrn4* (340 min) than in wt plants (100 min) (Figure 4D). Moreover, we ruled out the possibility that this is due to an increased activity of the *AFB2* promoter since the expression pattern of *AFB2p:GUS* was similar or lower, but not higher, in *xrn4* compared to wt plants (Appendix A). Thus, together these observations demonstrated that the increased accumulation of *AFB2* mRNAs in *xrn4*, and possibly that of other *TAAR* mRNAs, is directly due to improper decay of these mRNAs. Our experiments identified new targets of XRN4.

### 3.5. *xrn4* Plants Exhibit Hypersensitive Auxin Signaling Responses

We previously showed that the auxin-hypersensitive mutants *mir393a-1* and *mir393b-1* exhibit a decreased basal level of *DR5pro:GUS* expression and a conversely increased basal level of *IAA3pro:GUS*, of *IAA12pro:GUS,* and of *HS:AXR3-NT:GUS* expression, which we could explain by an increased basal level of AUX/IAA proteins [16]. To determine whether the auxin-dependent hypersensitive phenotypes of *xrn4* arise from similar hypersensitive auxin responses, we analyzed the expression of these reporter genes in *xrn4*. While the synthetic primary auxin-responsive gene *DR5pro:GUS* was expressed in the margins and in the hydathodes of leaves, and in the hypocotyls and in the root tips of wt plants, it was only weakly or not at all expressed in *xrn4* mutants (Figure 5A), and even after 2,4-D treatment, the induction of DR5pro:GUS was still weaker in *xrn4* mutant roots compare to the wt roots (Figure 5B). Thus, the hypersensitive auxin signaling responses recorded in *xrn4* were even stronger (Figure 5A) than those observed in *mir393a-1*, *mir393b-1,* and even *mir393ab* double mutant plants [16]. Conversely, the *IAA12pro:GUS,* which are not detected in wt plants, had much higher basal expression levels in emerging leaves and in the meristematic region of *xrn4* mutants (Figure 5D). In the case of IAA3pro:GUS, we did not observe a difference between *xrn4* and wt plants. However, like in miR393-deficient mutants, the expression of *IAA3pro:GUS* and *IAA12pro:GUS* was more rapidly induced in *xrn4* compared to wt plants by the synthetic auxin 2,4-D (Figure 5C,D). Thus, altogether, these results showed that, like *miR393* mutants, *xrn4* mutants exhibit strong hypersensitive responses to auxin.

To determine whether these hypersensitive auxin responses also rely on an increased basal level of AUX/IAA proteins, we analyzed the rate of AUX/IAA degradation by monitoring the accumulation of the AXR3-NT:GUS protein fusion (Figure 6) [16]. After induction of the *HSpro:AXR3-NT:GUS* gene for two hours at 37 °C (Figure 6), the AXR3-NT:GUS protein was stable in the roots of *xrn4* and accumulated to a level similar to that observed in *mir393b-1,* while it could not be detected in wt plants (Figure 6) [16]. Moreover, the two mutants occasionally exhibited ectopic expression of AXR3-NT:GUS in cotyledons and leaves (Figure 6). These observations showed that *xrn4-3* plants are also unable to properly degrade AUX/IAA proteins under normal growth conditions. To test whether these hypersensitive auxin responses were due to increased basal levels of AUX/IAA proteins as well, we treated the plants with 2,4-D for 4 h before inducing the *HSpro:AXR3-NT:GUS* gene. As we anticipated, chasing the high level of AUX/IAA proteins by inducing their degradation with high levels of auxin led to a restoration of the proper degradation of the AXR3-NT:GUS protein (Figure 6).

Thus, altogether, these experiments suggest that *xrn4*, like miR393-deficient plants, likely accumulate high basal levels of endogenous AUX/IAA proteins. More importantly, these experiments demonstrated that the hypersensitive auxin responses exhibited by *xrn4* are comparable and stronger than that exhibited by miR393-deficient mutants and that XRN4 is required to ensure the timely degradation of AUX/IAA proteins and to generate reliable auxin signaling outputs. 

## 4. Discussion

We have shown here that the 5′-3′ exoribonuclease XRN4 is a component of the auxin signaling pathway and that this function is achieved through a mechanism involving controlled degradation of TAAR transcripts. The Arabidopsis family of 5′-3′ exonuclease comprises 3 members, only one of which is localized in the cytoplasm and can potentially be involved in mature mRNA turn-over. XRN4 had been implicated in the degradation of 3′ RNA fragments generated upon microRNA-guided cleavages. Rymarquis and colleagues have shown that special features comprised within the 150 nucleotides downstream of the miRNA-binding site are necessary and sufficient for degradation of the 3′ cleavage fragments by XRN4 [25]. Their bioinformatic analyses also proposed that the basis for XRN4 selectivity might reside at least partially in the presence of hexameric nucleotide motifs. However, none of the xrn4-enriched transcripts used for the analysis have yet been shown to be directly degraded. It is therefore possible that these hexameric motifs rather represent sequences used by other proteins directly or indirectly regulated by XRN4. Similarly, the role of XRN4 in the ethylene signaling pathway occurs by indirectly controlling the mRNA level of EBF1 and EBF2. Because ethylene and auxin pathways are subjected to complex cross-talks at the level of their synthesis and likely at the level of their signaling, it is possible that the effects observed on EBFs are an indirect consequence of direct TAAR regulation. However, this hypothesis remains to be demonstrated. 

We have shown that the mRNAs of AFB2, and presumably those of the other TAARs, are directly degraded by XRN4. However, the basis for this selectivity or the underlying mechanism remain to be determined. Although the mechanism involved is unknown, we believe that it is highly regulated and specific for selected full-length transcripts. Indeed, accumulation of TAAR transcripts in the xrn4 mutants leads to hypersensitive responses to auxin and developmental defects. Studies by Gregory and colleagues have used deep sequencing approaches to identify Arabidopsis transcripts which might be decapped and degraded by XRN4 [36]. The experimental validation of at least some of these hypothetical targets will help determining the molecular determinants important for selection by XRN4. Of course, we also expect that the TAAR transcripts are decapped before degradation. Martinez de Alba and colleagues have shown that XRN4 is co-localized with decapping proteins such as decapping (DCP) and varicose (VCS) in the P-bodies [37]. Like in animals, where XRN1 interacts with DCP1, selection of TAARs for degradation by XRN4 should depend on a specific interaction with a decapping protein [38]. Martinez de Alba and colleagues have also shown that decapping prevents RDR6-dependent production of small RNAs. RDR6 is localized in siRNA bodies, which are themselves closely linked with the P-bodies, indicating a strong interaction between RNA turnover and RNA silencing. Thus, we can speculate that XRN4 could modulate the quantity of the matrix available for the production and amplification of siTAARs necessary for the TAARs silencing, and so to ensure an appropriate auxin response.

We anticipate that siTAARs may also act as non-cells autonomously. Indeed, the pathway that we have identified might be responsible for the generation of certain auxin signaling maxima important for initiation of lateral organ formation [39,40]. Indeed, considering that XRN4 would not be expressed in a given cell or cell group, this would lead to the accumulation of high levels of TAAR transcripts in these cells and to the production of high levels of siTAARs. These would create auxin signaling maxima by subsequently moving and silencing TAAR transcripts in the surrounding cells. Importantly, some of the phenotypic defects observed in XRN4, like the perturbed phyllotaxis characterized by the stochastic production of flowers on the stems or the high degree of leaf serration, strongly support our model [41] (Figure 1).

Our work reveals that TAAR mRNA are targets of XRN4 and demonstrates that their regulation by XRN4 is necessary for the proper auxin response and plant development.

## Figures and Tables

**Figure 1 genes-09-00638-f001:**
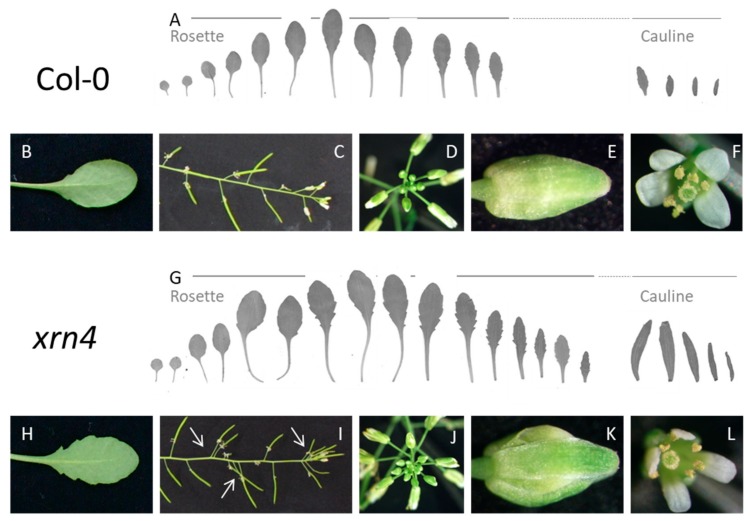
The *xrn4* mutants exhibit pleiotropic development defects. For Col-0 and *xrn4-3*, the photographs show representative series of rosette leaves and cauline leaves (**A**,**G**) and magnified views of the 5th leaves (**B**,**H**), stems with siliques and flowers (**C**,**I**), terminal inflorescences (**D**,**J**), closed flower buds (**E**,**K**), and opened flowers (**F**,**L**). See Table 1 for quantifications.

**Figure 2 genes-09-00638-f002:**
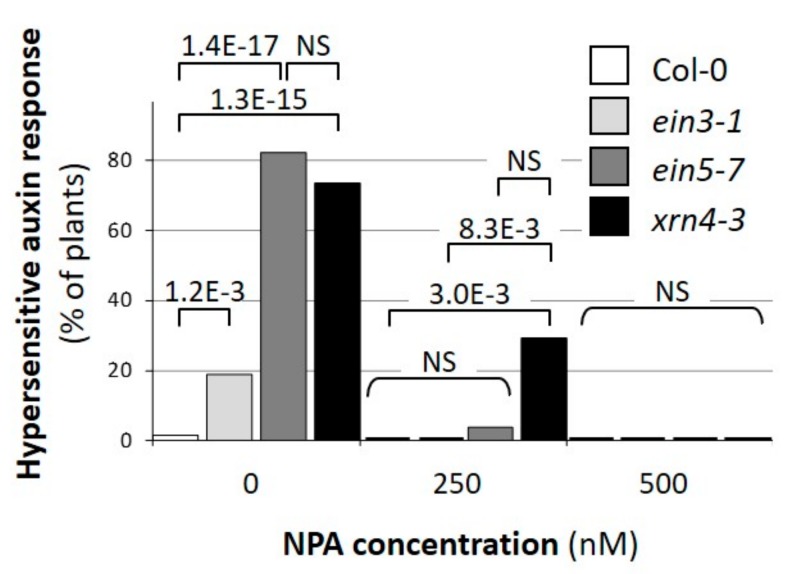
The *xrn4* mutants exhibit extreme auxin-dependent epinasty of cotyledons. The incidence of the auxin-hypersensitive response in populations of Col-0 (open bars), *ein3-1* (light grey bars), *ein5-7* (dark-grey bars), and *xrn4-3* mutants (dark bars). Seedlings (*n* > 17 for each condition and genotype) were grown on media containing the concentration of N-1-Naphthylphthalamic Acid (NPA) indicated and harvested four days after germination. The *p* values (two-tailed Fisher’s exact test) for significant differences towards each other are indicated.

**Figure 3 genes-09-00638-f003:**
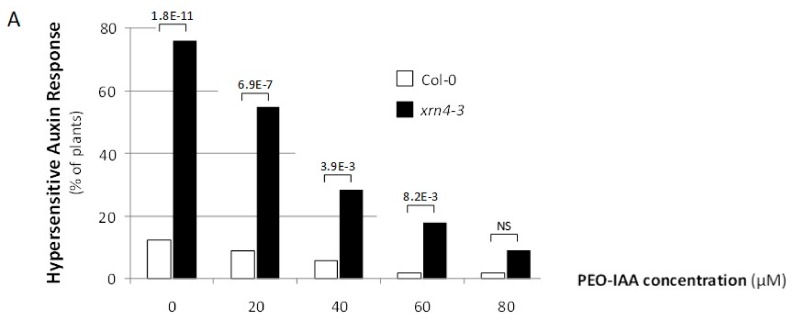
Reversion of *xrn4*’s developmental phenotypes by chemical suppression of TAAR functions. (**A**) The incidence of the auxin-hypersensitive response in populations of Col-0 (open bars); *xrn4-3* (solid bars) seedlings grown on media containing the concentration of the auxin antagonist PEO-IAA (alpha-(phenylethyl-2-oxo)-IAA) indicated. The *p* values (two-sided Fisher exact test) for significant differences between pairs is indicated; no significant (NS), *p* > 0.05. (**B**) The mean number of serrations per leaf in populations of Col-0 (open bars) and *xrn4-3* (solid bars) plantlets grown on media supplemented (+) or (−) with 80 μM of the auxin antagonist PEO-IAA. The *p*-values (Student t-test) for significant differences are indicated; NS, *p* > 0.05.

**Figure 4 genes-09-00638-f004:**
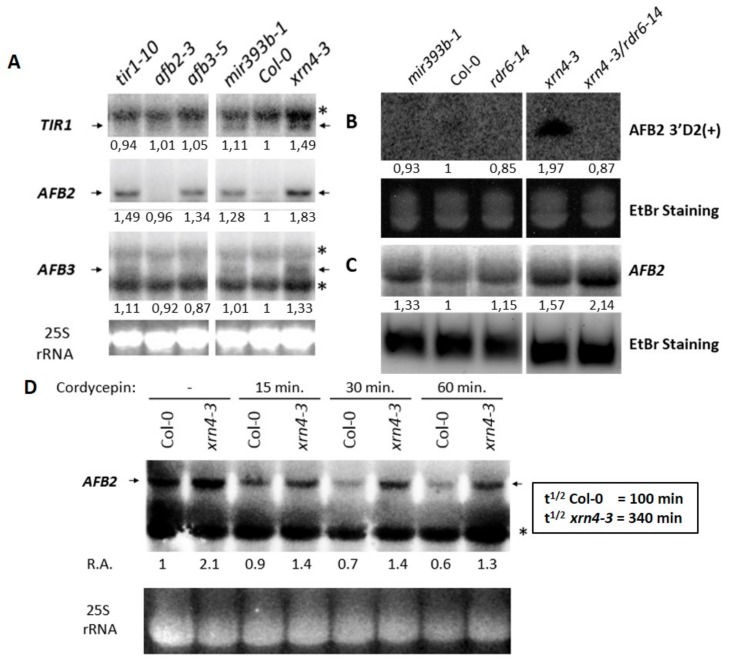
XRN4 is required for proper accumulation and turn-over of *TAAR* messenger RNAs (mRNAs) and siTAARs. (**A**) Comparison of *TIR1*, *AFB2,* and *AFB3* mRNA accumulation in Col-0, *xrn4-3,* and *mir393b-1* by Northern blot. Signals detected in *tir1-10*, *afb2-3,* and *afb3-5* mutants serve as negative controls; (**B**,**C**) analysis of AFB 3′D2(+) and *AFB2* mRNA accumulation, respectively, in *mir393b-1*, Col-0, *rdr6-14*, *xrn4-1,* and *xrn4-3/rdr6-14* by RNA-blot hybridization and Northern blot. (**D**) Analysis of *AFB2* mRNA decay by Northern blot in Col-0 and *xrn4-3* mutants for the time indicated after treatment with the transcription inhibitor cordycepin. The half-life (t1/2) of the *AFB2* transcript was determined by normalizing the *AFB2* signals to the 25S rRNA signals. Arrows indicate the full-length mRNA and asterisks indicate non-specific signals. *Midori Green*^®^ stained 25S rRNAs serve as a loading standard for northern blot and U6 for RNA-Blot hybridization.

**Figure 5 genes-09-00638-f005:**
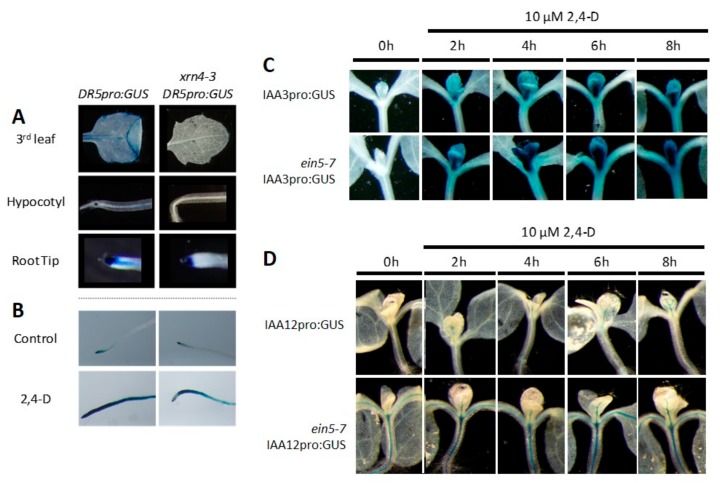
The *xrn4* mutant exhibits hypersensitive molecular responses to auxin. (**A**) Typical decrease in the basal expression levels of DR5pro:GUS in leaves, hypocotyl, and roots tips of *xrn4-3* compared to wt plants; (**B**) lant treatment with 2,4-D for 4 h induced the expression of DR5pro:GUS more rapidly in wt than in *xrn4-3* mutants; (**C**,**D**) expression levels of IAA3pro:GUS and IAA12pro:GUS in leaves of *ein5-7* compared to wt plants.

**Figure 6 genes-09-00638-f006:**
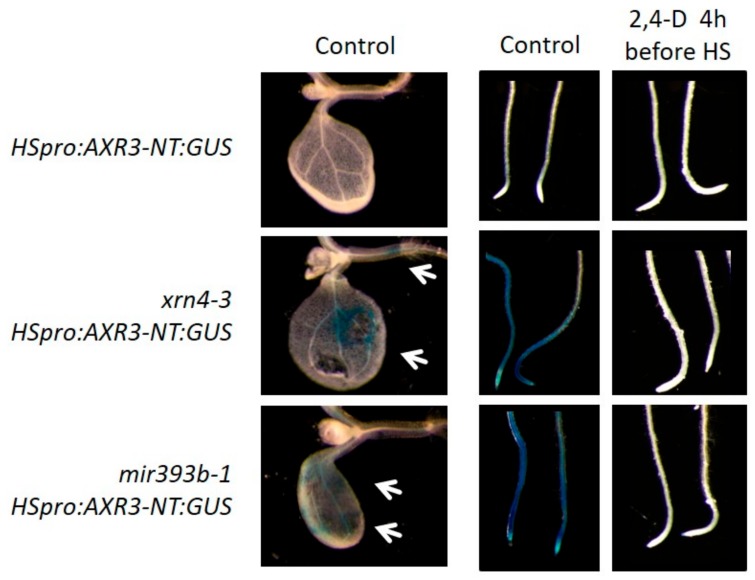
XRN4 is required for proper degradation of AXR3-NT:GUS proteins. Pictures of cotyledons and roots of wt, *xrn4-3* and *mir393b-1* mutant expressing the HSpro:AXR3-NT:GUS gene. Without treatments, the AXR3-NT:GUS fusion protein is more stable in *xrn4-3* and *mir393b-1* than in wild-type (wt) plants, and sometimes accumulates ectopically in cotyledons. Clearance of high levels of endogenous AUX/IAA proteins by treatment with 2,4-D for 4 h before heat shock (HS) allowed proper degradation of AXR3-NT:GUS.

**Table 1 genes-09-00638-t001:** Comparison of wild type (wt) and *xrn4* phenotypes at different stages of plant development.

	Col-0	*xrn4*	*p*-Value
**14 days-old**			
Primary Root Length (mm)	51.3 ± 1.3	61.6 ± 1.4	4.0 × 10^−6^
N. Lateral Roots	19.9 ± 1.2	12.8 ± 0.7	1.6 × 10^−5^
**28 days-old**			
Fresh Weight (mg)	105 ± 10	325 ± 29	4.4 × 10^−7^
N. of Serrations on 5th Leaf	0.8 ± 0.2	2.0 ± 0.2	3.1 × 10^−4^
N. of Serrations on 7th Leaf	1.3 ± 0.3	3.5 ± 0.2	2.5 × 10^−6^
**34 days-old**			
Number of Leaves	11.9 ± 0.2	17.4 ± 1.0	1.2 × 10^−5^
**45 days-old**			
N. of duplicated siliques	0.0 ± 0.0	3.3 ± 0.6	3.1 × 10^−4^
N. of aborted siliques	0.1 ± 0.1	0.7 ± 0.2	3.4 × 10^−2^
N. of Silique insertions	25.0 ± 1.3	29.3 ± 0.7	2.9 × 10^−3^
Internode length (mm)	7.9 ± 0.1	6.3 ± 0.2	2.7 × 10^−6^

The values are Mean ± SE (standard error) for N = 10 plants grown in standard long day (LD) conditions. Statistical *t*-test *p*-values relative to Col-0 are given. Root measurements were done on vertically grown plantlets. The number of leaves was determined at 34 days just before bolting of the plants. These data are for the *ein5-7* mutant allele. Similar values were recorded for the *ein5-1* allele.

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
