# Peer review of "The 5′-3′ Exoribonuclease XRN4 Regulates Auxin Response via the Degradation of Auxin Receptor Transcripts"

_genes, 2018, doi:10.3390/genes9120638_

Round 1
Reviewer 1 Report
The manuscript entitled: « The 5'-3' Exoribonuclease XRN4 regulates Auxin Response via the degradation of Auxin Receptor Transcripts » by Windels and Bucher reports the role of the RNA turnover machinery in auxin response.
The authors show that the XNR4 5’-3’ exoribonuclease participates in the degradation of the TIR1/AFB2 auxin receptor transcripts to ensure proper auxin response.
Combination of phenotypic analysis, genetics, molecular biology and histochemistry support the conclusions.
The manuscript is well written and the experiments perfectly designed.
These results add a new layer of knowledge in the regulation of auxin response.
Minor comments
Figures 4 A, B and C: I suggest to add quantification below each lane.
I do not see U6 hybridization in these figures but only in Fig S3. Could you please clarify.
Figure S2
I suggest to add p value for the time points showing significant differences
Author Response
Dear Reviewer,
Figures 4 A, B and C: I suggest to add quantification below each lane.
Like you suggested , we added quantification below each lane see new figure 4
I do not see U6 hybridization in these figures but only in Fig S3. Could you please clarify.
U6 is used only for control of smallRNA. For mRNA the Etbr staining is enough to show that the loading is equall.
Figure S2
I suggest to add p value for the time points showing significant differences.
It is true a p value it is necessary to show that the differences are significants. We added on the graph
Best regards
Reviewer 2 Report
Overall, interesting data and new piece of information regarding the regulation of the auxin signaling. Good methodological approach as well, figures are done in a nice and understandable way.
Regarding the style in which the manuscript is written; my understanding is that this is a follow up paper to the previously published work. However, written in its current form, it makes for a tough read to follow (it just cannot stand alone like this) because it is expected form the reader to be very familiar with all the previous work done. I suggest it to be rewritten in a way that the information relevant to understanding what and why was done is given. To give few examples; i) Figure 1 shows the phenotype of xrn4-3 mutants, but the Table1 is a summary of the phenotype of the allele ein5-7. You either present both for one allele or you explain somewhere in the text before the figure and the table that those two are allelic, i) in the Northern blots done for Figure 4, again, hard to follow why the levels of the AFB2 3'D2(+) were examined and why in the double mutant of xrn4-3/rdr6-14. Some information about that should be provided either in the introduction or explained in that specific result section.
Line 210, the DR5 pro:GUS show in the Figure 5A and the comment that it is higher than in mir393 mutants and the citation provided. Was the GUS staining on all those lines done together, are those experiments done in the same batch and thus comparable? If so, why not putting the image of the mir393 alongside the xrn4-3 like you did for Figure 6?
Sentence in the line 100 is not all that clear.
Author Response
Dear reviewer,
Regarding the style in which the manuscript is written; my understanding is that this is a follow up paper to the previously published work. However, written in its current form, it makes for a tough read to follow (it just cannot stand alone like this) because it is expected form the reader to be very familiar with all the previous work done. I suggest it to be rewritten in a way that the information relevant to understanding what and why was done is given. To give few examples; i) Figure 1 shows the phenotype of xrn4-3 mutants, but the Table1 is a summary of the phenotype of the allele ein5-7. You either present both for one allele or you explain somewhere in the text before the figure and the table that those two are allelic,
To answer we modified by this paragraph "We observed that mutants impaired in the function of XRN4 exhibit developmental abnormalities which resulted in an increase of their global stature and to alter the morphology of several of their organs like for the T-DNA insertion allele xrn4-3 (Figure 1). As exemplified in Table 1, ein5-7 mutants, another allele of xrn4, develop longer leaves and generate significantly more of these leaves (Student t-test, P=1.2E-5) than the wild-type (wt) plants. Their rosettes are three times heavier than those of wt plants (P=4.4E-7), their stems"
in the Northern blots done for Figure 4, again, hard to follow why the levels of the AFB2 3'D2(+) were examined and why in the double mutant of xrn4-3/rdr6-14. Some information about that should be provided either in the introduction or explained in that specific result section.
To answer we modified by this paragraph "To know if the increase of TAAR transcripts in xrn4 plants can affect the production of siTAARs derived from AFB2 (AFB2 3’D2(+)) (Fig. 4B), we analysed the accumulation of AFB2 3’D2(+) in xrn4-3 and xrn4-3/rdr6-14. Whereas the level of mir393 still remained unchanged in xrn4-3 (Fig. S3), we can observe an increase of AFB2 3’D2(+) in xrn4-3 but not in the double mutants xrn4/rdr6 like expected because the production of AFB2 3’D2(+) is dependent of RDR6."
Line 210, the DR5 pro:GUS show in the Figure 5A and the comment that it is higher than in mir393 mutants and the citation provided. Was the GUS staining on all those lines done together, are those experiments done in the same batch and thus comparable? If so, why not putting the image of the mir393 alongside the xrn4-3 like you did for Figure 6?
No the staining was not perform at the same time. For the comparison between mir393 mutants and xrn4 for DR5::GUS staining is clear because in xrn4-3 dr5::gus are not express at all but in mir393 mutants still some expression so it for that we wrote “Thus, the hypersensitive auxin signalling responses recorded in xrn4 were even stronger (Fig. 5A) than those observed in mir393a-1, mir393b-1 and even mir393ab double mutant plants”